# Characterization of the Filovirus-Resistant Cell Line SH-SY5Y Reveals Redundant Role of Cell Surface Entry Factors

**DOI:** 10.3390/v11030275

**Published:** 2019-03-19

**Authors:** Francisco J. Zapatero-Belinchón, Erik Dietzel, Olga Dolnik, Katinka Döhner, Rui Costa, Barbara Hertel, Barbora Veselkova, Jared Kirui, Anneke Klintworth, Michael P. Manns, Stefan Pöhlmann, Thomas Pietschmann, Thomas Krey, Sandra Ciesek, Gisa Gerold, Beate Sodeik, Stephan Becker, Thomas von Hahn

**Affiliations:** 1Department of Gastroenterology, Hepatology and Endocrinology, Hannover Medical School, 30625 Hannover, Germany; ZapateroBelinchon.Francisco@mh-hannover.de (F.J.Z.-B.); rui.costa@uk-essen.de (R.C.); bhertel@uni-potsdam.de (B.H.); A.Klintworth@gmx.de (A.K.); Manns.Michael@mh-hannover.de (M.P.M.); 2Institute for Molecular Biology, Hannover Medical School, 30625 Hannover, Germany; 3German Center for Infection Research (DZIF) Hannover-Braunschweig Site, Germany; Krey.Thomas@mh-hannover.de (T.K.); Sodeik.Beate@mh-hannover.de (B.S.); 4Institute of Virology, Philipps University Marburg, 35043 Marburg, Germany; erik.dietzel@staff.uni-marburg.de (E.D.); olga.dolnik@staff.uni-marburg.de (O.D.); becker@staff.uni-marburg.de (S.B.); 5German Center for Infection Research (DZIF) Gießen-Marburg-Langen Site, Germany; 6Institute of Virology, Hannover Medical School, 30625 Hannover, Germany; Doehner.Katinka@mh-hannover.de (K.D.); Veselkova.Barbora@mh-hannover.de (B.V.); 7Institute of Virology, Essen University Hospital, 45122 Essen, Germany; Sandra.Ciesek@uk-essen.de; 8TWINCORE, Center for Experimental and Clinical Infection Research, Institute for Experimental Virology, 30625 Hannover, Germany; jared.kirui@twincore.de (J.K.); Pietschmann.Thomas@mh-hannover.de (T.P.); gisa.gerold@twincore.de (G.G.); 9Infection Biology Unit, German Primate Center – Leibniz Institute for Primate Research, 37077 Göttingen, Germany; SPoehlmann@dpz.eu; 10Faculty of Biology and Psychology, University Göttingen, 37073 Göttingen, Germany; 11Department of Clinical Microbiology, Virology; Wallenberg Centre for Molecular Medicine (WCMM), Umeå University, 90187 Umeå, Sweden

**Keywords:** Filovirus cell entry, attachment factors redundancy, SH-SY5Y cell line, host–pathogen interactions

## Abstract

Filoviruses infect a wide range of cell types with the exception of lymphocytes. The intracellular proteins cathepsin B and L, two-pore channel 1 and 2, and bona fide receptor Niemann–Pick Disease C1 (NPC1) are essential for the endosomal phase of cell entry. However, earlier steps of filoviral infection remain poorly characterized. Numerous plasma membrane proteins have been implicated in attachment but it is still unclear which ones are sufficient for productive entry. To define a minimal set of host factors required for filoviral glycoprotein-driven cell entry, we screened twelve cell lines and identified the nonlymphocytic cell line SH-SY5Y to be specifically resistant to filovirus infection. Heterokaryons of SH-SY5Y cells fused to susceptible cells were susceptible to filoviruses, indicating that SH-SY5Y cells do not express a restriction factor but lack an enabling factor critical for filovirus entry. However, all tested cell lines expressed functional intracellular factors. Global gene expression profiling of known cell surface entry factors and protein expression levels of analyzed attachment factors did not reveal any correlation between susceptibility and expression of a specific host factor. Using binding assays with recombinant filovirus glycoprotein, we identified cell attachment as the step impaired in filovirus entry in SH-SY5Y cells. Individual overexpression of attachment factors T-cell immunoglobulin and mucin domain 1 (TIM-1), Axl, Mer, or dendritic cell-specific intercellular adhesion molecule-3-grabbing non-integrin (DC-SIGN) rendered SH-SY5Y cells susceptible to filovirus glycoprotein-driven transduction. Our study reveals that a lack of attachment factors limits filovirus entry and provides direct experimental support for a model of filoviral cell attachment where host factor usage at the cell surface is highly promiscuous.

## 1. Introduction

*Ebola* and *marburgviruses* are enveloped, negative single-strand RNA viruses of the *Filoviridae* family [1]. Since the discovery of Marburg virus (MARV) in 1967 [2] and Ebola virus (EBOV) in 1976 [3], the US Centre of Disease Control has reported several epidemic outbreaks in humans and nonhuman primates [4,5]. Despite intense world-wide research efforts, no antiviral treatments or vaccines have yet been licensed. In addition to primates, filoviruses infect pigs, dogs, duikers, and fruit bats in nature, and rodents and ferrets can be infected experimentally [6,7,8,9,10,11,12].

The viral glycoprotein (GP), the only viral surface protein, exclusively mediates the entry and internalization of filoviruses into cells. The precursor protein GP0 is synthesized on the endoplasmic reticulum, and cleaved in the constitutive secretory pathway into the surface unit GP1, which binds to host cell factors, and the transmembrane unit GP2, which mediates fusion of viral envelopes with endosomal membranes. Filoviruses display a broad cell tropism [13]. Almost any cell type with the notable exception of lymphocytes is susceptible to infection by authentic filoviruses in vitro [14,15], or to transduction by retrovirus particles pseudotyped with GP [16,17]. Moreover, immortalized cell lines cultured in suspension are resistant to filovirus entry, while cell adhesion enhances susceptibility to infection [18,19]. Thus, the broad cell tropism observed in infected primates, where virus can be isolated from all organs but not from lymphocytes [14,20,21], is also recapitulated in vitro.

The availability of host factors on the cell surface that interact with viral envelope GP or with envelope lipids such as phosphatidylserine (PtdSer) often determines viral cell tropism. Such virus–host interactions mediate virus attachment, and are a necessary prerequisite for virus internalization, viral fusion with host membranes, and viral genome release into the cytosol for transcription and replication [16,22,23]. Several plasma membrane proteins have been implicated in filovirus attachment: cellular lectins such as asialoglycoprotein receptor (ASGR-R), dendritic cell-specific intercellular adhesion molecule-3-grabbing non-integrin (DC-SIGN), liver/lymph node-specific intercellular adhesion molecule-3-grabbing non-integrin (L-SIGN), human macrophage C-type lectin specific for galactose and N-acetylglucosamine (hMGL), or liver and lymph node sinusoidal endothelial cell C-type lectin (LSECtin) [24,25,26,27,28], T-cell immunoglobulin and mucin domain 1 and 4 (TIM-1, TIM-4) [29,30], members of the TAM family (Tyro3, Axl, Mer) of receptor tyrosine kinases [31], integrin αVβ1 [32,33], and scavenger receptor A. However, none of these factors seems to be essential for filoviral infection across cell lines. Rather, their role in cell entry is considered to be cell type dependent, and some of them may promote entry indirectly by regulating downstream processes such as macropinocytosis or GP proteolytic cleavage [34,35,36,37]. In contrast, several intracellular proteins are essential for filovirus infection in all cell types studied thus far. The endosomal and lysosomal cysteine proteases cathepsin B and cathepsin L cleave GP and thereby expose its receptor binding domain [38], and the two-pore channel 1 (TPC1) and two-pore channel 2 (TPC2) mediate endolysosomal Ca^2+^ efflux [39]. Finally, the endolysosomal cholesterol transporter Niemann–Pick C1 (NPC1) [40,41] binds to processed GP1 [42].

The remarkable diversity of plasma membrane proteins implicated in filovirus cell entry prompted us to analyze twelve cell lines for a potential correlation of host factor expression to filovirus susceptibility. We could show that the neuroblastoma SH-SY5Y cell line is specifically resistant to filovirus infection although all intracellular proteins known to be essential were expressed, and although its overall transcriptome was very similar to that of susceptible cell lines. Heterokaryon assays revealed that SH-SY5Y cells did not express a dominant restriction factor that inhibited filovirus GP-driven cell entry, but recombinant GP could not bind to their plasma membrane. By individual overexpression of a wide range of different filovirus attachment-promoting proteins, SH-SY5Y cells became susceptible to filovirus GP-driven transduction. Our findings demonstrate that filoviruses—in contrast to many other viruses—can highjack a diverse range of different plasma membrane proteins for the critical cell attachment step, which is a mandatory prerequisite for productive infection of any cell and thus any host.

## 2. Material and Methods

### 2.1. Cell Lines

HepG2, Huh-7.5, SW13, HEK293T, EA.hy926, A549, and 786-O cells were a kind gift from Charles M. Rice (Rockefeller University, NY, USA). HPMEC and hTERT-BJ1 cells were generously provided by Andreas Tiede (Department of Hematology, Hemostasis, Oncology, and Stem Cell Transplantation, Hannover Medical School, Hannover, Germany). SK-N-BE(2)-C, SK-N-MC, and SH-SY5Y were kindly provided by Herbert Hildebrant (Department of Cellular Chemistry, Hannover Medical School, Hannover, Germany). Adherent immortalized cell lines were cultured in Dulbecco´s Modified Eagle Medium (DMEM) supplemented with 10% Fetal Calf Serum (FCS) (Sigma-Aldrich, San Luis, MS, USA), penicillin/streptomycin (100 U/mL), nonessential amino acids, and L-glutamine (2 mM) at 37 °C and 5% CO_2_. The suspension cell line Jurkat was a kind gift of Abel Viejo-Borbolla (Institute of Virology, Hannover Medical School, Hannover) and was cultured with Roswell Park Memorial Institute (RPMI) 1640 medium plus 25 mM HEPES with the same supplements and conditions as adherent cell lines. Primary human hepatocytes (PHH) were provided by Florian Vondran (Department of Surgery, Hannover Medical School, Hannover).

The cell lines Huh-7.5/Tet3G and SH-SY5Y/Tet3G were engineered to stably express Tet-On 3G transactivator protein, and the cell line HEK293T/ZsGreen1 was developed to stably express ZsGreen1 fluorescent reporter protein under the control of Tet-On 3G-inducible P_TREG3V_ promoter. Genetically modified cell lines were generated by stable transduction of parental cell lines either with lentiviral particles of the pLVX-Tet3G plasmid vector (Clontech) for transactivator expression or pLVX-TRE3G-ZsGreen1 for reporter expression. 0.6 mg/mL G418 or 2 µg/mL puromycin were added in the media of transduced cell lines to select for transactivator or reporter expressing cells, respectively. Clone HEK293T-H6/ZsGreen1 was isolated by cell subcloning to obtain a permanent cell line with homogenous reporter expression. Stable cell lines were maintained in culture as parental cell lines with 10% Tet-Free FBS (Clontech, Takara Bio, Mountain View, CA, USA) in instead of FCS.

### 2.2. DNA Plasmid Constructs

Surrogate HIV-based pseudovirion packaging plasmid HIV Gag-pol, transfer plasmids CSLucW2 (Firefly luciferase expression) or CSGW (hrGFP), and envelope plasmid for VSV-G protein and its backbone plasmid pcDNA3.1 have been described previously [43]. The pWPI-mCherry-BSD bicistronic expression vector was generated by introduction of the mCherry DNA sequence into a pWPI-BSD empty vector [44]. Both plasmids were digested with restriction enzymes BamHI and SpeI. The mCherry encoding fragment was gel isolated, purified and ligated into the new lentivirus expressing vector. mCherry expression was confirmed by pseudovirion generation and fluorescence-activated cell sorting (FACS) analysis. Lenti-X^TM^ Tet-On^®^ inducible expression system plasmids pLVX-Tet3G and pLVX-TRE3G-ZsGreen1 were purchased from Takara Bio. (Mountain View, CA, USA).

### 2.3. Ebolavirus

GP encoding plasmids pVR1012-GP(Z) (EBOV Mayinga), pVR1012-GP(R) (RESTV Pennsylvania), pVR1012-GP(S) (SUDV Boniface), and pVR1012-GP(IC) (TAFV Cote d´Ivoire) were provided by Drs. Anthony Sanchez and Gary Nabel (NHI, Bethesda, USA). pCAGGS-MARV, pCAGGS-BDBV or pCAGGS-LLOV encoding for MARV Musoke BDBV Bundibugyo and *Cuevavirus* LLOV were a gift from Drs. Heinz Feldmann (MARV and BDBV) (NIH, Hamilton, USA) and Ayato Takada (Research Center for Zoonosis Control, Hokkaido, Japan), respectively. EBOV Makona Kissidougou-C15 and mutant variant A82V GP expression plasmids were a donation of Jonathan K. Ball (School of Life Science, University, Nottingham, UK). pCAGGS-SARS-S expressing spike protein S of SARS-CoV strain Frankfurt has previously been described [45]. pCMV-LassaGPC (expressing LASV strain AV GP precursor) was provided by Dr. Francois-Loic Cosset (INSERM U758, Lyon, France). pMD.RVG.CVS24-B2c encoding for the GP of the rabies virus variant B2c (RABV B2c) was a gift from Manfred Schubert (Addgene plasmid # 19713; http://n2t.net/addgene:19713; RRID:Addgene_19713). pIRES-EGFP-CHIKV E3-E1 was a kind gift of Dr. Barbara Schnierle (Department of Virology, Paul-Elrich-Institut, Langen, Germany). EBOV Mayinga GP1-human Fc fusion protein expression plasmid pAB61-ZEBOV-GP-Fc and pAB61 are published [46].

Overexpressing vectors for tyrosine Axl (pLV[Exp]-Neo-EF1A>hAXL[NM_021913.4]), Mer (pLV[Exp]-Neo-EF1A>hMERTK[NM_006343.2]) or NPC1 (pLV[Exp]-Bsd-EF1A>hNPC1[NM_000271.4]*) were commercially purchased (VectorBuilder Inc, Shenandoah, United States). MLV-based DC-SIGN overexpressing plasmid pQCXIP_hDC-SIGN-AU1 has been described [47]. For construction of pWPI-HAVCR1-BSD vector, the coding sequence of human TIM-1 (hTIM-1) was synthesized (Integrated DNA Technologies) and amplified by PCR using the forward primer 5’CCTGCAGGCGCGCCGGATCCGCCACC ATGCATCCTC AAGTGGTCAT CTTAAGCCTC ATCCTACATC TGGCAGATTC TGTAGCTGGTTCTGTAAAGG TTGGTGG3′ and the reverse primer 5′GATATCCGGAGCCGCCTCCTCCATCCGTGGCATAAAGACTATTC3′ and cloned into the lentiviral expression vector pWPI-BSD [44] by Gibson assembly. The hTIM-1 insert was confirmed by sequencing.

### 2.4. Reagents and Antibodies

The drugs Z-Phe-Tyr(tBu)-diazomethylketone (FYdmk) (219427, Calbiochem, San Diego, CA, USA), U18666A (662015, Calbiochem, San Diego, CA, USA) and tetrandrine (sc-201492, Santa Cruz Biotechnology, Santa Cruz, CA, USA) were dissolved in DMSO according to supplier’s instructions. Working solutions were prepared by dilution of the stock solution with culture medium. Doxycycline hyclate (DOX) (D9891, Sigma-Aldrich, San Luis, MS, USA) for cell–cell fusion reporter expression activation was prepared as 50 mg/mL stock solution in DMSO and stored at −80 °C. The stock solution was diluted in culture media supplemented with Tet-free approved FBS (Clontech, Takara Bio, Mountain View, CA, USA) to 500 ng/mL. Recombinant conjugated protein EGF-AlexaFluor555 for EGF intravesicular accumulation analysis was purchased from Invitrogen™ (E35350) (Carlsbad, CA, USA). Filipin III (F4767) for unesterified cholesterol detection was obtained from Sigma-Aldrich (San Luis, MS, USA).

Mouse monoclonal allophycocyanin(APC)-conjugated antibodies (mAbs) against human surface entry factors Axl (FAB154A), human integrin β1 (FAB17783A), Mer (FAB8912A), or DC-SIGN (FAB161A) and their respective isotype controls, IgG_1_ (IC002A) and IgG_2B_ (IC0041A) as well as AlexaFluor488-conjugated mAb against human integrin αV (FAB1219G-025) and its IgG_1_ isotype control (IC002G) were purchased from R&D systems (Minneapolis, MN, USA). Mouse APC mAb against human TIM-1 (353905) or APC-mIgG_1_ κ isotype control (400121) were obtained from Biolegend Inc. (San Diego, CA, USA). For immunoblot detection of intracellular factors rabbit polyclonal antibodies (pAbs) directed against human procathepsin L + cathepsin L ab (ab200738) or mouse antihuman Niemann Pick-C1 mAb (ab55706, ab134113) were purchased from Abcam (Cambridge, UK). Antihuman procathepsin B and cathepsin B rabbit mAb (31718) was acquired from Cell Signaling Technology Inc. (Danvers, MA, USA). Rabbit pAbs directed against human two pore channel segment 1 (TPC1) (LS-C110014) was purchased from LifeSpan Bioscience Inc. (Seattle, WA, USA) or against human two pore channel segment 2 (TPC2) (ACC-072) from Alomone labs (Jerusalem, Israel). Mouse mAb against human β-tubulin (T7816) or GAPDH (G9545) were purchased from Sigma-Aldrich (San Luis, MS, USA). Horseradish peroxidase-coupled (HRP) goat pAbs directed against mouse (A4416) or against rabbit (A6154) IgGs were purchased from Sigma-Aldrich (San Luis, MS, USA). Mouse pAb conjugated with Fluorescein isothiocyanate (FITC) and directed against human IgG, Fc_γ_ fragment specific ab (209-095-098) for fusion protein cell binding assays was purchased from Jackson ImmunoResearch Europe Ltd. (Newmarket, UK).

### 2.5. Pseudovirion Production and Transduction

HIV-based pseudotype viral particles were generated as described previously [43]. In brief, 90% HEK293T cells were cotransfected in 6-well plates with 1 µg total DNA using polyethylenimine (PEI) (Sigma-Aldrich, San Luis, MS, USA) according to manufacturer’s instructions. DNA ratios were 1 part transfer plasmid:1 part packaging plasmid:4 parts envelope plasmid for reporter pseudotype virus generation or 7:7:1 for transgene delivery. Cell supernatants were collected at 48 and 72 h post-transfection and passed through a 0.45 µm pore filter. Viral stocks were stored at 4 °C for short time (less than 7 days) or aliquoted and frozen at −80 °C for long-term storage.

For transduction assays, viral supernatants were first mixed with polybrene (H9268-5G, Sigma-Aldrich, San Luis, MS, USA) at a final concentration of 4 µg/mL and 500 µL or 1 mL was added to 30–50% confluent target cells (80–90% for heterokaryon transductions) in 12 or 6-well plate format, respectively, for 6 h. Seventy-two hours post transduction, cells were lysed for luciferase quantification or processed for fluorescence detection. For virus–cell specificity studies, GFP-based pseudoparticles were titrated by limiting serial dilution and subsequent transduction of the susceptible cell line Huh-7.5. Transducing units per ml (TU/mL) were calculated as [((% infected cells/100) * # transduced cells) / virus volume (mL)] by FACS analysis. For pharmacological inhibition studies, target cells were preincubated with drugs in DMSO or DMSO alone for 1 h prior to addition of pseudoparticles and further incubation.

Luciferase-based GP-driven entry was measured as previously described [48]. For filovirus GP-driven susceptibility cell screening, nonspecific (NoEnvpp), EBOV (EBOVpp), or MARV (MARVpp) GP-mediated entry relative light units (RLU) values were normalized to positive control VSV GP-driven (VSV-Gpp) values for cross-experimental comparison [(EBOVpp or MARVpp RLUs / VSV-Gpp RLUs) × 100] and displayed as luciferase activity arbitrary units. EBOV species susceptibility evaluation is expressed as absolute RLU values in 100 µl cell lysate. For fluorescence reporter expression, target cells were quantified by FACS and shown as the % positive cells from total.

### 2.6. Authentic Filovirus Infection

Fully infectious filovirus work was performed under Biosafety Level 4 (BSL4) conditions at the Institute of Virology, Phillips-Marburg University, Germany. Susceptible cell lines SK-N-BE(2)-C and HEK293T and resistant cell lines SK-N-MC and SH-SY5Y seeded on 6-well plates with 1 coverslip per well were mock treated, or infected with EBOV (Mayinga isolate) (GenBank accession number NC002549) or MARV (Musoke isolate) (GenBank accession number NC001608) at a MOI of 0.1 TCID50/mL for 1 h at 37°C. After inoculation, virus was removed, and cells were further incubated for 72 h in DMEM with 3% FCS. Three days later, the cells were fixed with 4% PFA for 48 h. For immunofluorescence analysis, free aldehyde groups were quenched with 0.1 mM glycine in PBS, and the cells were permeabilized with 1% Triton-X 100 (TX100). Permeabilized cells were incubated with goat α-EBOV or α-MARV antisera (α-EBOV serum was used for mock infection) against virus Nucleoprotein (NP) as primary Ab. α-goat AlexaFluor488 was used as secondary Ab with DAPI for nuclear staining. NP staining was visualized using a Zeiss Axiophot microscope with a 40x objective with an illumination of 1000 ms for FITC channel and 500 ms for DAPI.

Remnant cells were scraped from the wells into 1% SDS/PBS for immunoblot analysis. 3% lysates were run through an SDS/PAGE gel and transferred onto a nitrocellulose membrane. NP or tubulin protein detection was carried out by the same goat EBOV and MARV antisera as used for IF and a tubulin specific mab, respectively, followed by antigoat IRDye680-conjugated secondary ab incubation. Immuno-labelled proteins were detected with and odyssey infrared-imaging system (LI-COR).

### 2.7. rVSVΔG-EBOVGP Infection

Briefly, HEK293T and SH-SY5Y cells were infected with 500 µL/well of recombinant VSV bearing EBOV Mayinga strain GP (rVSVΔG-EBOVGP) at a MOI of 1 (3.3 × 10^5^ TCID50) or mock infected for 1 h at 37 °C in DMEM without FCS. At 1 hpi, 2.5 mL DMEM with 3% FCS was added to the virus inoculum and further incubated for 24 h (2 replicates) or 48 h (1 replicate). Cell supernatant was collected each day, and viral titers were determined by TCID50 limiting-dilution assay 48–72 h after starting of assay [49]. Simultaneously, bright-field images were acquired at 24 h (2 replicates) and 48 hpi (1 replicate) with a Nikon TS100 microscope using a 10× objective.

### 2.8. Flow Cytometry

For cell surface protein detection immunostaining and subsequent fluorescence quantification was performed. In brief, adherent cells were detached with versene (15040066, Gibco, Gaithersburg, MD, USA), a nonenzymatic cell dissociation reagent. The cell suspensions were spun at 700 rpm for 5 min, and the cells were resuspended in 200 µL 2% FCS in PBS (FACS buffer) solution per staining. Fc receptors were blocked with 1 µL human FcR blocking reagent (Miltenyi Biotec Gmbh, Bergisch Gladbach, Germany) for 10 min. Subsequently, saturating concentrations of conjugated specific or isotype control Abs were added and incubated for at least 15 min. Stained cells were washed with FACS buffer and resuspended in 400 µL FACS buffer for fluorescence detection. Cells were kept at 4 °C and in the dark throughout the staining. Protein expression was evaluated by APC or FITC channel intensity in a BD FACSCanto cytometer (BD, Frankin Lakes, NJ, USA). Protein expression was calculated as the difference of geometric mean fluorescence intensity (ΔMFI) between specific Ab and isotype control staining. Cell surface detection of bound GP1-Fc fusion protein was detected as described before [50]. Briefly, adherent cells were detached with versene. 5 × 10^5^ cells/well were washed with 1× DPBS containing Ca^2+^ and Mg^2+^ (14040133, Gibco, Gaithersburg, MD, USA) and Fc receptors blocked with human FcR blocking reagent. Subsequently, cells were incubated with 100 nM GP1-Fc or Fc for 1.5 h, washed with FACS Buffer^++^ (2% FCS in PBS with Ca^2+^/Mg^2+^) and incubated with FITC-conjugated mouse Abs directed against human Fc fragment ab (1:200) for 45 min. Cells were washed twice with FACS Buffer^++^ and fixed in 1 part FACS Buffer^++^/3 parts 3% PFA for 10 min. Cells were maintained at 4 °C throughout the processing. Bound protein was quantified by FITC channel intensity in a BD FACSCanto cytometer (BD, Frankin Lakes, NJ, USA) and subtracted from MFI of secondary Ab control. To monitor GFP, cells were detached with trypsin, washed with 1× PBS and fixed with 3% PFA for 10 min at 4 °C and analyzed with a BD FACSCanto cytometer. Cell–cell fusion/transduction fluorescence quantification analysis was performed at the Hannover Medical School Cell Sorting Core Facility with a BD FACSAria Fusion (BD, Frankin Lakes, NJ, USA). Cells were prepared as described for GFP quantification. The gating strategy for heterokaryon detection and transduction is depicted in Appendix A. 10,000 events were recorded for single cell line controls, 50,000 for the PBS treatment control, and 10,000 ZsGreen1 positive events (heterokaryon cells) for PEG treatment. Data was analyzed with FlowJo7 software (Tree Star, Ashland, OR, USA).

### 2.9. Expression of EBOV GP1-Fc Fusion Protein

Plasmids encoding soluble EBOV Gp1-Fc or Fc alone were transfected by polyethylenimine into HEK293Tcells. Cell supernatants with EBOV GP1-Fc and Fc, respectively, were collected 48 and 72 h post transfection, concentrated using a tangential flow cassette (Vivaflow 200) (Sartorius, Gottingen, Germany) and diluted with an equal amount of 20 mM phosphate buffer (pH 7). Proteins were purified by affinity chromatography using a HiTrap Protein G HP 1 mL column (GE Healthcare, Braunschweig, Germany) according to the manufacturer’s instructions followed by concentration and size exclusion chromatography using a Superose 6 Increase 10/300 GL column (GE Healthcare, Braunschweig, Germany) equilibrated with 10 mM Tris pH 8, 150 mM NaCl.

### 2.10. Western Blot

To detect intracellular filovirus entry factors immunoblotting was conducted. Cells were lysed in Cell Lytic M (C2978, Sigma-Aldrich, San Luis, MS, USA) plus protease inhibitors (11836145001, Roche, Basel, Switzerland). Protein concentrations were determined by Bradford assay (23200, Thermo Fischer Scientific, Waltham, MA, USA) using a Nanodrop™ 2000 spectrophotometer (ND-2000, Thermo Fischer Scientific, Waltham, MA, USA). 1.5 µg of protein was solubilized with 6× DTT sample buffer (Tris-HCl, Glycerin, SDS, DTT, Bromophenol), boiled for 5 min at 95 °C, cooled down and resolved by electrophoresis. Proteins were blotted to a PVDF membrane (GE Healthcare, Braunschweig, Germany) by Mini Trans-Blot^®^ (Bio Rad, Hercules, CA, USA) wet transfer system. Membranes were blocked and incubated with ab in 5% milk powder with 0.1% PBS-Tween20 either for 1 h at RT or o/n at 4 °C with constant shaking. Protein was detected using the HRP-based ECL chemiluminescence system (GE Healthcare, Braunschweig, Germany) and blots were documented with a LAS-4000 Mini (Fujifilm, Düsseldorf, Germany).

### 2.11. RNA Extraction and Microarray Analysis

Cells were grown to about 90% confluency on 10 cm dishes prior to extraction. Total RNA was extracted as instructed in the RNeasy Midi kit (Qiagen, Hilden, Germany) guidelines.

Microarray analysis was carried out at the Research Core Unit Transcriptomics of Hannover Medical School. The whole Human Genome Oligo Microarray Kit 4x44K v2 (G4845A, design ID 026652, Agilent Technologies, Santa Clara, CA, USA), covering 44495 probes and about 26,000 transcripts, was used to analyze mRNA levels. Quick Amp Labeling Kit, two-color (5190-0444, Agilent Technologies, Santa Clara, CA, United States) was used for the generation of Cy3- or Cy5- labelled cRNA. Raw data was acquired as Relative Light Units (RLU). Microarray data has been deposited in public genomic data repository Gene Expression Omnibus (GEO) under the accession number GSE104008.

### 2.12. Hierarchical Clustering Analysis (HCA)

Microarray data was compiled for gene expression clustering analysis among studied cell lines. Briefly, matrices were prepared by assigning attributing individual expression values of each gene (row) versus the corresponding cell line (column). Gene probes that yielded no detectable signal in any of the studied cell lines were excluded from the analysis. An unbiased heatmap was generated from these matrixes through the R software heatmpap.2 gplot package [51] by applying the “euclidean” distance method and the “average” (UPGMA) clustering method. Color scales represent values between the maximum and minimum expression values of the individual cell lines.

### 2.13. Cathepsin B and L Activity Assay

Cysteine proteases cathepsin B and L proteolytic activities were measured using the fluorometric cathepsin L (ab65300) and cathepsin B (ab65306) activity assay kit (Abcam, Cambridge, UK) according to the manufacturer´s protocol with the modification of the starting material. 1 × 10^6^ cells were spun down, washed and lysed in 150 µl chilled CL lysis buffer. Lysates were split into 3 wells with equal volumes (≈3.3 × 10^5^ cells/well) for intra-assay variation evaluation. Fluorometric values were recorded with a microplate reader using a Bio-tek FLx 800 Multifunction Fluorescence Luminescence Microplate Reader (Bio-Tek, Winooski, USA) with a 528/20 emission filter.

### 2.14. Cholesterol and EGF Endosomal Accumulation Assays

NPC1 and TPC1/2 functionality was evaluated by unesterified cholesterol and EGF endolysosomal accumulation, respectively. 8 × 10^4^ HEK293T or 2 × 10^5^ SH-SY5Y cells were seeded onto poly-L-lysine coated coverslips in a 24-well plate. On the next day, cells were treated with vehicle (DMSO) control or target-specific drugs for 24 h. Twenty-four hours post-treatment, cells for NPC1-mediated cholesterol shuttling analysis were washed and fixed with 3% PFA in PBS for 20 min at RT. Furthermore, cells for TPC1/2 functionality assessment were incubated for 30 min with 1 µg/mL EGF-AlexaFluor555 conjugated protein in serum-free DMEM without or with 2 µg/mL tetrandrine. Subsequently, cells were washed with serum-free DMEM and treatments applied for extra 3.5 h in normal culture medium. Finally, cells were washed and fixed as for cholesterol accumulation.

### 2.15. Light Microscopy

Fluorescence microscopy was used to assess NPC1 or TPC1/2 functionality or heterokaryon formation and the susceptibility of heterokaryons to filovirus GP-mediated transduction. Sample preparation was performed as described previously [52]. For functionality assays, cells were fixed with 3% paraformaldehyde in PBS for 20 min followed by quenching of the remaining fixative with 50 mM NH_4_Cl for 10 min and permeabilization with 0.1% TX100 for exactly 5 min. The cells were washed in PBS and blocked with 0.5% BSA in PBS for 30 min. To assess the functionality of NPC1 or TPC1/2, samples were then stained with 0.5% BSA/PBS solution containing 50 µg/mL Filipin III (unesterified cholesterol) or 0.05 mg/mL DAPI, respectively, for 30 min followed by extensive washing. All incubations were done at RT. Lastly, coverslips were washed with H_2_O and mounted with Mowiol containing 2.5% (wt/vol) 1,4-diazabicyclo-[2.2.2]octane on glass slides. For cell–cell fusion/transduction assays, coverslips were processed identically as for TPC1/2 samples. EGF accumulation (TPC1/2) and heterokaryon formation and transduction were analyzed at a confocal fluorescence microscope (TCS SP8, Leica microsystems, Wetzlar, Germany) with plan-apochromat 63×/1.40 oil immersion objectives, and 405-, 488-, 561-, and 633-nm lasers. Cholesterol accumulation (NPC1) was analyzed at a Leica DM5000 B epifluorescence microscope (Leica microsystems, Wetzlar, Germany) coupled to a UV light filter. Pictures were acquired with a 40x magnification objective. Documentation was performed with LAS AF Lite software (Leica microsystems, Wetzlar, Germany) and image processing with ImageJ2/Fiji package [53] or Adobe Photoshop CS4 (Adobe Systems Inc., Mountain View, CA, USA).

### 2.16. Polyethylene Glycol (PEG) Mediated Cell–Cell Fusion

Cells expressing either DOX-inducible Tet-On 3G transactivator (Huh-7.5, SH-SY5Y) or ZsGreen1 under the control of the P_TREG3V_ promoter (HEK293T-H6) were chemically fused. Cell–cell fusion methodology has been previously described [54]. 5 × 10^5^ Huh-7.5 or 5 × 10^5^ SH-SY5Y cells were coseeded with 3 × 10^5^ HEK293T-H6 cells in 6-well plates. Twenty-four hours after coculture, the cells had reached about 80–90% confluency increasing the chance of cell–cell contacts and hence fusion efficiency. Cocultured cells were either treated with 500 µl PBS as fusion control or with pre-warmed 40% PEG1500 (10783641001 Roche, Sigma-Aldrich, San Luis, MS, USA) as fusogenic agent for 5 min at 37 °C. After incubation, any trace of PEG was removed by extensive washing with PBS. Finally, cells were allowed to recover for at least 1 h at 37 °C with fresh new media prior to further experimentation.

### 2.17. Statistical Analysis

Unless stated otherwise, experiments were carried out as 3 biological replicates with 3 technical replicates for intra-assay control and quantified as the mean of the 9 data points with error bars representing standard deviation (SD). Viral transductions were performed with 3 different viral stocks. Microscopy experiments were performed twice for functionality assays. Cell–cell fusion/filovirus and transduction was documented once by microscopy and quantified by flow cytometry in three independent experiments. Data analysis and representation was done with Graph Pad Prism 7 statistical software (La Jolla, CA, USA). Multiple *t* tests was performed for statistical significance discovery correcting for multiple comparison with the Holm–Sidak method. *P*-value significance was represented as: n.s. *P* > 0.05; * *P* ≤ 0.05; ** *P* ≤ 0.01; *** *P* ≤ 0.001; **** *P* ≤ 0.0001

## 3. Results

### 3.1. SH-SY5Y and SK-N-MC Cells Are Resistant to Filovirus GP-Driven Lentiviral Transduction

To identify potential cell-type differences in filovirus susceptibility, we transduced a panel of twelve cell lines with HIV-based particles coding for the reporter luciferase and pseudotyped with the EBOV strain Mayinga GP (EBOVpp), the MARV strain Musoke GP (MARVpp), the VSV-G Indiana strain G (VSV-Gpp), or lacking any viral GP (NoEnvpp). At 72 h post transduction, we measured the luciferase activity and normalized it for each cell line to the one obtained upon VSV-Gpp transduction (Figure 1A,B). Particles bearing no GP did not result in any reporter expression, while VSV-G particles lead to strong luciferase activity in all cell lines. Susceptibility to filovirus GP-driven transduction was heterogeneous with the endothelial cell lines HPMEC and EA.hy926 being much more susceptible than the fibroblasts and epithelial cells, and the lymphocytic cell line Jurkat being resistant, as reported before [16]. Remarkably, the adherent, neuroblastoma cell lines SK-N-MC and SH-SY5Y [55,56] were also not susceptible to filovirus GP-mediated transduction. Due to this unexpected phenotype, we set out to investigate the determinants underlying the observed resistance.

### 3.2. rVSVΔG-EBOV-GP and Authentic Filovirus Cannot Infect SH-SY5Y Cells

To validate our results obtained with lentiviral pseudotypes, we infected the different cell lines with the replication-competent VSV virus bearing the EBOV GP (rVSVΔG-EBOV-GP) or with authentic filoviruses. We inoculated the susceptible HEK293T and SK-N-BE(2)-C cells as well as the resistant SK-N-MC and SH-SY5Y cells with EBOV (Mayinga) or MARV (Musoke) at a multiplicity of infection (MOI) of 0.1 pfu/cell for 1 h. At 72 h post infection (hpi), the cells were fixed for immunofluorescence microscopy (IF) studies (Figure 2A) or lysed for immunoblotting (Figure 2B), and probed with antibodies (pAbs) directed against the viral nucleoprotein (NP). The HEK293T and SK-N-BE(2)-C cells were susceptible to EBOV and MARV infection, while the SH-SY5Y cells were resistant to both (Figure 2A,B). Of note, tubulin was also degraded in the susceptible SK-N-BE(2)-C cells but not in the resistant SH-SY5Y upon inoculation, a sign of extensive infection previously observed in our lab. The SK-N-MC did not express any EBOV NP protein, but displayed sporadic MARV NP labelling in few cells. Therefore, we focused our further analyses on SH-SY5Y cells.

Next, we inoculated HEK293T or SH-SY5Y cells with rVSVΔG-EBOVGP virus at a MOI of 1. We evaluated cell rounding and detachment, an established phenotype of EBOV GP-expressing cells [57], and measured the titers of released virus at 24 hpi. In line with our other experiments, the HEK293T cells had initiated cell rounding at 24 hpi (Appendix A), and completed it at 48 hpi (Figure 2C), while the morphology of the SH-SY5Y cells had not changed upon inoculation by 24 h (Appendix A) or 48 h (Figure 2C). Furthermore, viral titers had increased by 51-fold in HEK293T but decreased in SH-SY5Y by 24 hpi (Figure 2D). Altogether, these experiments demonstrated that SH-SY5Y cells were resistant to filovirus infection.

### 3.3. SH-SY5Y Cells Can Be Transduced by Many Viral Envelope Proteins but Not by GP of Filoviruses

The genus *Ebolavirus* comprises five species: *Zaire ebolavirus* (Ebola virus (EBOV)), *Sudan ebolavirus* (Sudan virus (SUDV)), *Reston ebolavirus* (Reston virus (RESTV)), *Bundibugyo ebolavirus* (Bundibugyo virus (BDBV)), and *Taï Forest ebolavirus* (Taï Forest virus (TAFV)) [1]. Moreover, a new filovirus species, *Lloviu cuevavirus* (Lloviu virus (LLOV)), was discovered in Spain in 2011 [58]. We asked whether the GPs of these viruses could mediate productive entry in SH-SY5Y cells. To test this, we transduced either HEK293T or SH-SY5Y cells with pseudoparticles bearing no viral envelope protein (NoEnvpp) or GPs from the respective *Ebola* species or *Lloviu cuevavirus*. For EBOV, we used GPs of 1976 Mayinga strain, 2014 Makona strain (WT), and the cell entry-enhancing Makona variant A82V [59,60]. VSV-Gpp served as positive control (Figure 3A). Consistent with our other results, HEK293T cells but not the SH-SY5Y cells were susceptible to transduction mediated by all GPs (Figure 3A).

To determine whether SH-SY5Y cells were susceptible to transduction driven by other viral GPs, we generated pseudoparticles encoding for GFP and bearing Chikungunya virus envelope proteins (CHIKV E1-E3), severe acute respiratory syndrome virus S protein (SARS-S), Lassa virus GPC (LASV-GPC), Rabies virus G protein (RABV-G), or VSV-G GPs (Figure 3B). Filovirus susceptible Huh-7.5 and HEK293T cells (Figure 1A–B) as well as resistant SH-SY5Y and Jurkat cells were transduced at a MOI of 0.1. While Huh-7.5 cells were susceptible to all pseudoparticles tested, Jurkat cells were effectively transduced only with VSV-Gpp. HEK293T cells were susceptible to all pseudoparticles with the exception of SARSpp, which could only transduce Huh-7.5 cells. LASVpp efficiently transduced epithelial Huh-7.5 and HEK293T cells. SH-SY5Y cells were susceptible not only to VSV-Gpp, but also to transduction driven by RABV-G and CHIKV E1-E3. In addition, marginal transduction was detected with LASV-GPC pseudoparticles. In summary, SH-SY5Y cells are resistant to transduction driven by filovirus GPs but not by the majority of the other viral glycoproteins tested.

### 3.4. SH-SY5Y Cells Do Not Express a Dominant Restriction Factor

Continuous virus–host co-evolution has led to the emergence of host restriction factors that limit many infections, and of viral proteins that counteract these intrinsic defense mechanisms. For instance, HIV-1 has evolved proteins that target the restriction factors APOBEC3G, TRIM5, or SERINC5 (reviewed in [61]). Similarly, EBOV GP antagonizes the antiviral protein tetherin [62].

To investigate whether the resistance of SH-SY5Y cells to GP-driven transduction was due to the expression of a host restriction factor, we used doxycycline (Dox)-controlled transcriptional activation [63], and combined it with experimental cell–cell fusion assays [54]. This combination allowed us to differentiate between heterokaryons of two different cell types expressing either the transactivator or its inducible counterpart from homokaryons derived from only one cell type. In the latter case, the inducible gene would not be activated, and hence there would be no reporter gene expression even in the presence of Dox (Appendix A). We engineered Huh-7.5 and SH-SY5Y cells to express transactivator proteins, and HEK293T-H6 cells to express the Tet-inducible ZsGreen1 reporter gene. We next assessed whether NoEnvpp, EBOVpp, MARVpp, or VSV-Gpp lentiviral particles expressing the mCherry reporter were able to transduce HEK293T-H6 cells, Huh-7.5 cells, SH-SY5Y cells, HEK293T-H6/Huh-7.5 heterokaryons, or HEK293T-H6/SH-SY5Y heterokaryons. The engineered cell lines exhibited the same susceptibility to GP-driven transduction as the respective parental cell lines (Figure 4B and Appendix A). Moreover, heterokaryons of susceptible HEK293T and Huh-7.5 cells or of HEK293T and SH-SY5Y cells were susceptible to GP-driven transduction (Figure 4A,B and Appendix A). Importantly, reporter expression depended on viral GPs, since NoEnvpp did not transduce heterokaryons. Taken together, heterokaryons from resistant and susceptible cells were transduced by pseudoparticles bearing filovirus GPs, suggesting that SH-SY5Y cells do not express a specific restriction factor but rather lack an important enabling filovirus entry factor, which is delivered to resistant cells by fusion with susceptible cells.

### 3.5. Intracellular Filovirus Entry Factors are Expressed and Functional in SH-SY5Y Cells

To further investigate the resistance phenotype of SH-SY5Y cells, we analyzed protein expression (Figure 5A,C,E) and functionality (Figure 5B,D,F) of the intracellular proteins operating during the endolysosomal phase of filovirus infection [39,40,41]. Cathepsin L was expressed to a similar extend in all cell lines, while expression of procathepsin B (44 kDa) and the active cathepsin B isoforms (27 kDa and 24 kDa) varied among cell lines but did not correlate with susceptibility to filovirus infection (Figure 5A). For instance, SH-SY5Y cells expressed the 27 kDa active isoform at a level comparable to that of susceptible HPMEC cells. In addition, we measured substrate-specific cleavage activity of cathepsin B and cathepsin L in the cell lysates (Figure 5B), but the relative cleavage activities did not correlate with susceptibility to filovirus GP-driven transduction. Notably, the resistant SH-SY5Y cells showed cathepsin B and cathepsin L substrate-specific cleavage activity, suggesting that GP-processing can occur on this cell line. Finally, specific inhibition of cathepsin B/cathepsin L with FYdmk [64] confirmed that filovirus GP-driven transduction into susceptible cell lines required cathepsin B and cathepsin L activity as expected (Appendix A).

The bona fide receptor NPC1 was ubiquitously expressed but its relative expression did not correlate with filovirus susceptibility (Figure 5C). For instance, HPMEC, the cell line with the highest susceptibility to EBOV or MARV GP-mediated transduction (Figure 1A,B), displayed the lowest NPC1 expression. We also assessed NPC1 functionality in HEK293T or SH-SY5Y cells by inhibiting its cholesterol transport activity with U18666A [65] (Figure 5D). In untreated cells, cholesterol was distributed throughout the entire cells with the highest signal at the plasma membrane. In contrast and as reported before [66,67], cholesterol accumulated more in the cell center in U18666A treated cells, most likely in late endosomes and endolysosomes, indicating that these cells expressed functional NPC1. Finally, U18666A treatment inhibited filovirus GP-driven transduction of susceptible cells (Appendix A), confirming that entry into these cells is NPC1 dependent. In sum, the data show that SH-SY5Y cells express functional NPC1.

Similarly, all cell lines expressed also the endosome traffic regulators TPC1 and TPC2 (Figure 5E). As for the other intracellular host factors, TPC1 relative expression was heterogeneous and did not correlate with susceptibility to transduction driven by the GPs of EBOV or MARV. On the other hand, TPC2 was expressed to a similar extent with SH-SY5Y cells displaying the lowest level of TPC2. To test whether TPC2 accounted for the resistance of SH-SY5Y cells to GP-driven transduction, susceptible HEK293T and resistant SH-SY5Y cells were treated with the inhibitor tetrandrine, which blocks TPC1/2 and results in epidermal growth factor (EGF) accumulation in LAMP-1 positive endosomes [39]. Both cell lines displayed larger and brighter organelles after drug treatment, indicating EGF accumulation and suggesting that also TPC1/2 were functional in SH-SY5Y cells. Finally, tetrandrine inhibited filovirus GP-driven transduction of all susceptible cell lines tested as expected (Appendix A).

Collectively, these results confirmed the essential functions of cathepsin B, cathepsin L, NPC1, TPC1, and TPC2 for filovirus infection, and demonstrated that neither an absence nor a dysfunction of these intracellular proteins can explain the resistance of SH-SY5Y cells to GP-mediated transduction or infection.

### 3.6. Filovirus Resistance Is Neither Explained by SH-SY5Y Cells’ Transcriptome nor by the Expression Profile of Filovirus Attachment Factors

Besides these intracellular proteins essential for filovirus infection, the resistance of SH-SY5Y could also result from a lack of plasma membrane proteins implicated in filovirus entry. Since a long list of host factors seems to contribute to filovirus attachment (Appendix A), we performed a global gene profiling analysis. We extracted total RNA from our panel of cell lines as well as from primary human hepatocytes (PHH) and quantified relative mRNA levels by microarrays. We used hierarchical clustering analyses to compare the respective gene expression profiles. Figure 6A represents an unbiased clustering analysis of all detected transcripts and Figure 6B depicts clustering of all filovirus cell surface or soluble extracellular entry factors described so far. The cell lines were grouped in two major clusters by an unbiased analysis (Figure 6A). This observation also hold true for the analysis considering only reported entry factors (Figure 6B), except that the Huh-7.5 cells were outside the cluster closest to PHH. Whereas one cluster contained most of the susceptible cell lines, the other enclosed all resistant cell lines, including SH-SY5Y, together with the susceptible HEK293T, SW13, and Huh-7.5 cell lines (unbiased analysis only). Thus, the gene expression profiles of the susceptible cell lines HEK293T and SW13 were more similar to resistant cell lines than to other susceptible lines. Moreover, neither the global gene expression profiles nor expression patterns of reported filovirus attachment factors correlated with susceptibility or resistance to filovirus GP-driven cell entry.

### 3.7. Surface Expression of Plasma Membrane Filovirus Entry Factors Does Not Correlate with Susceptibility

We next analyzed the microarray data in more detail, as the clustering analysis did not reveal any correlation between susceptibility to filovirus infection and host gene expression. We considered gene expression values below 150 Relative Light Units (RLU) on a specific probe as likely negative, 150 ˂ RLU ˂ 300 as uncertain, and above 300 as likely positive (Appendix A). mRNA levels that scored consistently above (ex. *TYRO3*) or below (ex. *CD209*) the threshold in all cell lines were not further considered. Of the remaining, *HAVCR1* (TIM-1) and *AXL* (Axl) mRNA levels were below the threshold in resistant SH-SY5Y and Jurkat cells but above in some susceptible cell lines. Therefore, we use FACS to analyze their surface expression (Figure 7), as well as of integrin αV and β1 (Appendix A), which have been implicated in filovirus entry too [32,33], and whose mRNA were less abundant in resistant than in susceptible cell lines (Appendix A).

Most cell lines had detectable Axl expression at the cell surface (Figure 7A,B). Among the susceptible cell lines, we observed both high levels (HPMEC or EA.hy926) and low levels (Huh-7.5, HEK293T). Conversely, TIM-1 was expressed only in three susceptible cell lines (Huh-7.5, A549, and 786-O) (Figure 7C,D). Integrins αV and β1 were present in all cells with some susceptible cell lines expressing similar levels to resistant ones (Appendix A). Notably, resistant SH-SY5Y cells lacked or expressed only low levels of all tested plasma membrane proteins.

In summary, the presence or absence of a specific cell attachment factor previously implicated in filovirus entry does not explain the resistance of the SH-SY5Y cells to filovirus infection.

### 3.8. Attachment Limits GP-Driven Entry into SH-SY5Y Cells

To further characterize the step of filovirus infection impaired in SH-SY5Y cells, we studied filovirus GP-specific attachment using recombinant soluble EBOV GP1 fused to human Fc or a human Fc alone [24] (Figure 8A). We chose HepG2 cells as a control, since they express the GP-binding C-type lectin ASGP-R and bind to filovirus GP [28,68], and compared them with susceptible HEK293T and resistant SH-SY5Y cells. EBOV GP1-Fc bound most efficiently to HepG2 cells and to HEK293T cells with lower efficiency. However, it did not bind to SH-SY5Y cells, suggesting that the latter lacks any attachment factor for EBOV GP.

### 3.9. Diverse Cell Surface Factors Can Overcome the Block to Filovirus GP-Driven Cell Entry in SH-SY5Y Cells

We next asked whether ectopic expression of surface factors that promote filovirus attachment could rescue GP-driven entry into SH-SY5Y cells (Figure 8B). We chose to overexpress Axl and TIM-1 proteins because their expression was either very low or absent in SH-SY5Y cells, but substantial in a subset of susceptible cell lines (Figure 7A–D, Appendix A). Furthermore, we overexpressed Mer tyrosine kinase since it was expressed well in HEK293T and SW13 susceptible cell lines, which had clustered in the transcriptome analysis with the resistant cell lines (Figure 6B, Appendix A). Finally, we overexpressed DC-SIGN as one Ca^2+^-dependent C-type lectin and NPC1 as negative control. We could confirm expression and correct subcellular localization of these attachment proteins (Appendix A) in the respective SH-SY5Y derived cell lines.

Next, we transduced the novel overexpressing SH-SY5Y cell lines, SH-SY5Y/Empty Vector, SH-SY5Y wild type (WT), or HEK293T with mock, EBOVpp, MARVpp, VSV-Gpp, or NoEnvpp lentiviral particles coding for firefly luciferase. Strikingly, ectopic expression of any of the four attachment protein tested was sufficient for robust transduction of SH-SY5Y cells by both, EBOVpp or MARVpp (Figure 8B), but not by NoEnvpp. As expected, NPC1 overexpression did not increase filovirus susceptibility, supporting the notion that filovirus attachment proteins had been missing. Among the ectopically expressed proteins, DC-SIGN increased EBOV GP-driven transduction by 364-fold as compared to SH-SY5Y WT. TIM-1, Axl, and Mer had a less potent yet significant effect on EBOV GP-mediated entry with fold change values of 66.7, 9, or 3.5, respectively. The transduction by MARVpp was stronger than by EBOVpp for both Axl (14.1) and Mer (10.2) tyrosine kinases, but not for TIM-1 (54.4). Moreover, DC-SIGN expression augmented MARVpp transduction by 15-fold, but this was not statistically significant. Finally, PtdSer receptors Mer and TIM-1 enhanced VSV-Gpp transduction by approx. 3-fold, as reported before [23,69]. However, Axl failed to promote transduction.

These results indicate that attachment was the limiting step of filovirus GP-driven entry into SH-SY5Y cells, and that filoviruses can utilize multiple alternative surface proteins for successful attachment to cells.

## 4. Discussion

The large number of host proteins reported to promote filoviral internalization and thus infection prompted us to analyze systematically the cellular requirements for filovirus susceptibility. We tested a panel of immortalized cell lines for EBOV and MARV GP-mediated transduction to correlate susceptibility to the presence of plasma membrane proteins or endosomal proteins required during the early phases of infection. As previously reported, fibroblast, epithelial, and endothelial cells were susceptible to filovirus GP-driven transduction, whereas the lymphocytic cell line Jurkat was fully resistant to pseudoparticle mediated reporter expression [16,22]. However, we identified two neuroblastoma cell lines, SK-N-MC and SH-SY5Y, which were resistant to filovirus GP-driven transduction. This was unexpected, as most human cell lines previously described to be resistant to filovirus infection are nonadherent or lymphocytic cell lines [14,16,19].

SK-N-MC was isolated from metastatic neuroblastoma tissue of a 14 years old Caucasian female [56]. These cells were not susceptible to EBOV-GP transduction or EBOV infection but supported a low level of MARV-GP transduction and MARV infection. This disparity might be due to different receptor usage among filoviruses, to different internalization mechanisms, or even indicate a host factor specifically required for EBOV but not for MARV [17]. On the other hand, SH-SY5Y cells, a thrice cloned subline of the neuroblastic SK-N-SH cell line [56,70], were highly resistant to both EBOV and MARV by all assays (Figure 1 and Figure 2). These cells are susceptible to chikungunya virus, dengue virus, varicella-zoster virus, and neurovirulent poliovirus [71,72,73,74]. Our study revealed that SH-SY5Y cells are also susceptible to transduction driven by the G protein of rabies virus (Figue2B). Together, these results suggest that the entry defect observed is largely filovirus specific.

According to the current model on filovirus cell entry, one or several proteins on the cell surface facilitate infection in addition to the essential intracellular endosomal proteins. At the plasma membrane, Ca^2+^-dependent carbohydrate binding C-type lectins interact with N-glycans attached to filoviral GP [75] and function as attachment factors [34]. Furthermore, members of the TAM (Tyro3, Axl, Mer) and TIM-1 PtdSer binding protein families enhance filovirus cell entry, TAMs likely by enhancing macropinocytosis [37] and TIM-1 by a process depending on GP or PtdSer [30,76]. Finally, the intracellular proteins cathepsin B and cathepsin L, TPC1 and TPC2, and the bona fide receptor NPC1 are essential for productive filovirus infection in vitro and in vivo [38,40,41]. Our panel of cell lines was largely lacking expression of lectins at the mRNA level with the exception of ASGR1 and ASGR2, which were expressed on Huh-7.5 cells, as reported previously [28]. The absence of expression of other entry promoting lectins is in line with the tissue distribution of DC-SIGN, hMGL (dendritic cells and macrophages), L-SIGN, and LSEctin (endothelial cells of the liver and lymph nodes) [24,25,27]. Therefore, although lectins might contribute to the cell tropism of filoviruses [20], clearly they are not essential for filovirus infection. In line with previous studies [30,31,36,68,77], TAM and TIM-1 mRNA expression was heterogeneous among the cell lines studied here, suggesting that they are not essential but may facilitate filovirus attachment and internalization. Indeed, studies on the role of Axl in filovirus cell entry have shown a cell-type-specific effect with antibodies against Axl reducing Axl-dependent filoviral GP-driven transduction in some Axl-positive cell lines but not in others [77]. Similarly, overexpression of DC-SIGN but not of TIM-1 renders Jurkat cells susceptible to filovirus virus-like particles (VLP) [76] and infection studies with TIM-1^-/-^ mice also suggest a dispensable role of TIM-1 since EBOV viral loads are similar in KO and WT mice [78]. Our data clearly demonstrate that TIM-1, Axl, Mer, and DC-SIGN have important yet redundant roles during filovirus entry. This is in line with published studies supporting the function of C-type lectins, TAM tyrosine kinases, and TIM proteins in cell entry [23,24,27,30,31,69,79,80,81]. Prospective further attachment factor protein expression profiling among susceptible and resistant cell lines could provide even a clearer picture on the cellular requirements for productive filovirus cell entry. In sum, expression of TIM-1, Axl, Mer, or DC-SIGN is an important determinant of susceptibility to filoviruses.

Finally, the required intracellular endosomal proteins were expressed and functional in all cell lines studied here including the resistant SH-SY5Y cells, irrespective of their susceptibility to EBOV infection, suggesting that they are not the determinants of resistance in SH-SY5Y cells. A similar scenario has been observed in resistant lymphocytic cell lines, which clearly express NPC1 yet they are resistant to infection [68]. In fact, the Jurkat cells analyzed here had higher *NPC1* mRNA levels than primary human hepatocytes (432 over 308 RLU) (Appendix A) and expressed NPC1 protein (Figure 5B). Thus, resistance to filoviral cell entry is more common than previously thought and not restricted to either nonadherent or lymphocytic cells or to the presence of intracellular host factors.

The disparity between susceptibility to infection and surface protein expression could be attributed to different reasons. First, resistant cell lines might express a dominant restriction factor that blocks productive internalization or viral fusion (reviewed in [61]). However, this was not the case for the SH-SY5Y cells reported here, as heterokaryons of susceptible HEK293T-H6 and SH-SY5Y cells were also susceptible to filovirus GP-driven transduction. Second, filoviruses might rely on different cofactors in different cell types. In Jurkat cells, this might be conceivable. An overexpression of DC-SIGN but not of TIM-1 renders otherwise resistant Jurkat cells susceptible to EBOV-GP virus-like particles [76]. Moreover, EBOV virions bind to resistant Jurkat and activated CD4+ T-cells in a TIM-1 dependent-manner without productive infection [78]. However, ectopic expression of Axl, an inhibitor of the type I interferon signaling [69], renders Jurkat cells susceptible. This suggests that Jurkat cells might be able to resist filovirus infection because they have an intrinsic antiviral innate immune mechanism activated. This hypothesis could be tested in the future by determining whether heterokaryons of resistant Jurkats with susceptible cells will be susceptible or resistant to filovirus infection. In SH-SY5Y cells, however, an ectopic expression of any potential attachment-promoting plasma membrane protein of filoviruses, such as DC-SIGN, Axl, Mer, or TIM-1, conferred susceptibility to filovirus GP-driven transduction or infection. Interestingly, the enabling potency was different among attachment factors. These differences may be related to the mechanism by which the attachment factors promote susceptibility in SH-SY5Y cells. For instance, TAM receptors may confer susceptibility by enhancing micropinocytosis, whereas TIM-1 or DC-SIGN could bind to the virus GP [26,30,77]. Further mechanistic studies on the role of the studied attachment factors could shed light on the differences observed in GP-mediated filovirus susceptibility in SH-SY5Y cells. Nevertheless, these data highlight the concept that virus–receptor interactions at the cell surface are required for productive entry but several alternative host cell factors can be utilized. To our knowledge, this is the first study which directly shows that several unrelated proteins can be utilized alternatively to initiate filoviral cell entry in one cellular context. These findings may help to explain the broad cell tropism of filoviruses.

In summary, we have characterized SH-SY5Y cells as a novel adherent nonlymphocytic cell line exhibiting a specific resistance to filoviral cell entry, and show that GP-mediated filovirus attachment to cells is impaired. This lack of susceptibility could be circumvented by expression of any one of several unrelated cell surface proteins that had previously been implicated in filoviral cell entry using other cellular backgrounds and experimental settings. We thus provide the most direct experimental evidence to support the current model of filoviral cell entry with alternative unspecific interactions between filoviral particles and plasma membrane host proteins mediating attachment and subsequent infection of target cells. Moreover, the filovirus-specific resistance of SH-SY5Y cells makes this cell line an interesting new tool for studying filoviral cell entry and for potentially identifying new cellular factors in the filoviral entry puzzle.

## Figures and Tables

**Figure 1 viruses-11-00275-f001:**
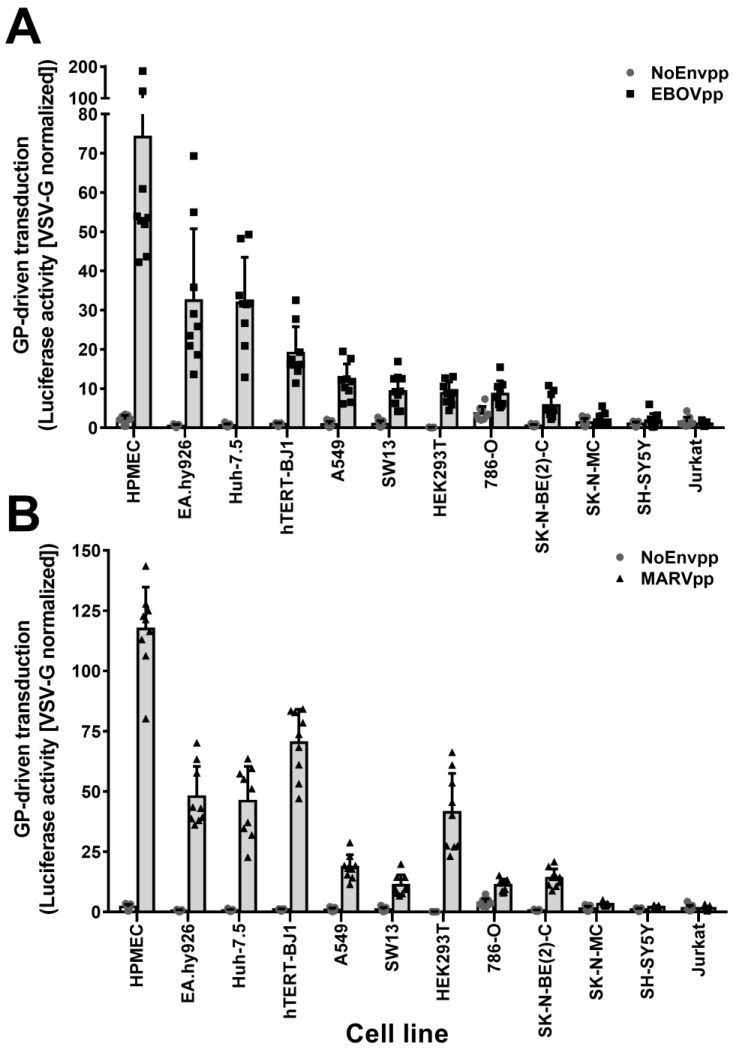
Unbiased cell screening for susceptibility to EBOV (EBOVpp) and MARV (MARVpp) GP-driven transduction. A panel of twelve cell lines were transduced with EBOV (**A**) or MARV (**B**) firefly luciferase pseudoparticles. Pseudoparticles with no envelope proteins (NoEnvpp) or VSV-G (VSV-Gpp) were used as negative and positive control, respectively. Seventy-two hours post transduction, cells were lysed and luciferase activity was measured. Data were normalized to VSV-Gpp activity as indicated in material and methods. Graphs plot the mean values of three independent experiments performed in triplicate (*n* = 9) with error bars representing the standard deviation (SD). Individual values are represented as dots, squares or triangles.

**Figure 2 viruses-11-00275-f002:**
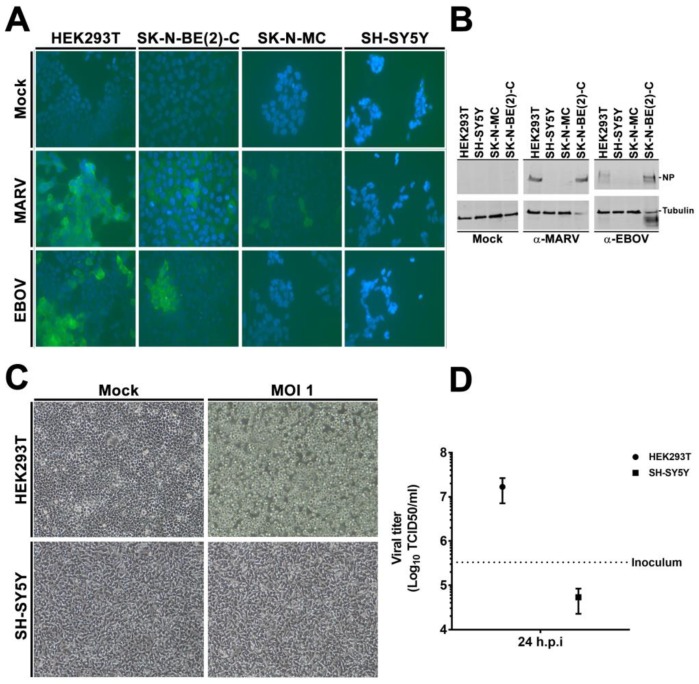
SH-SY5Y is resistant to EBOV and VSVΔG-EBOVGP infection. (**A**,**B**) Authentic filovirus infection. Susceptible HEK293T, SK-N-BE(2)-C as well as resistant SK-N-MC and SH-SY5Y cells were infected with EBOV or MARV or mock-infected for 1 h at a MOI of 0.1. Seventy-two hpi, cells were fixed and stained for EBOV and MARV NP (**A**) using DAPI as counterstaining. Images acquired with a 40× objective (**B**) In parallel, remaining cells were lysed, and analyzed by immune-blotting for virus NP. α-tubulin was used as internal control. Images and blots are representative of three independent infections. (**C**,**D**) rVSVΔG-EBOV infection. HEK293T and SH-SY5Y cells were infected with rVSV bearing EBOV Mayinga GP for 1 h at 37 ºC. Fresh media was added and cells further incubated for 48 h. A 10× objective was used for image acquisition (**C**) 48 hpi. GP-specific cell rounding and detachment. (**D**) Cell supernatant was collected 24 hpi and viral titers determined by TCID50.

**Figure 3 viruses-11-00275-f003:**
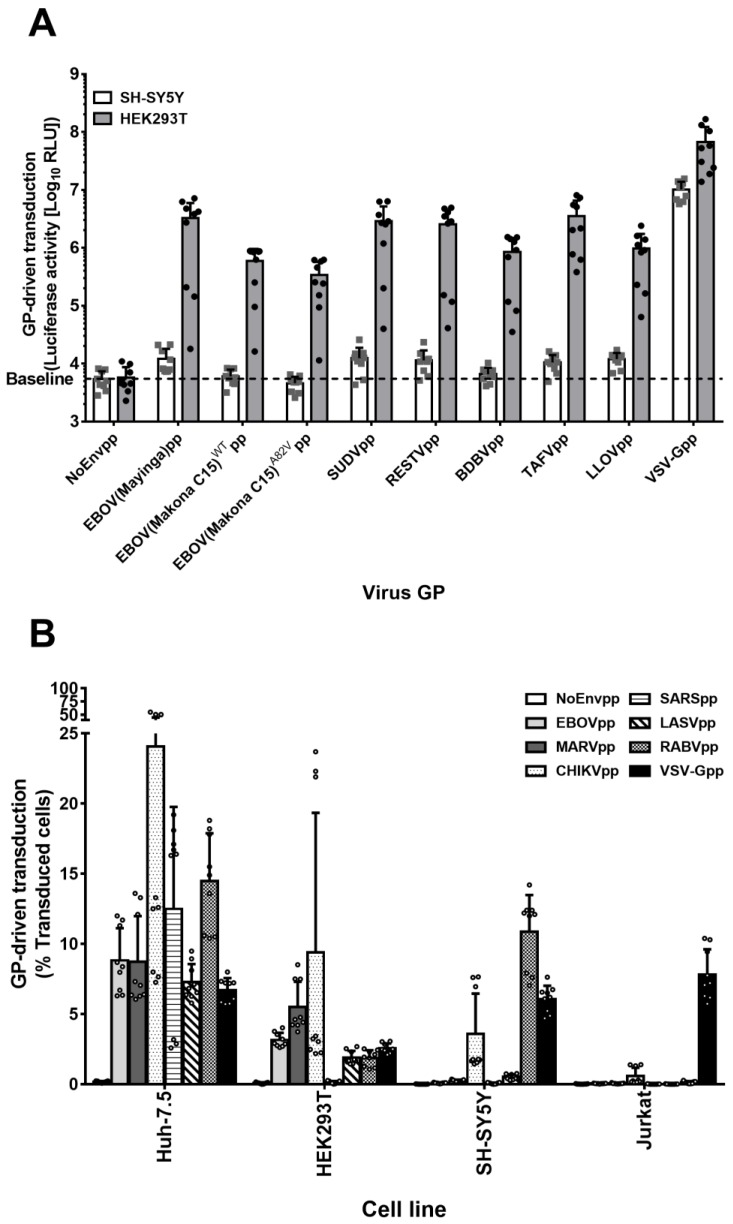
SH-SY5Y cells are resistant to pseudoparticles bearing either Ebolavirus species GPs but susceptible to pseudoparticles bearing glycoproteins of other virus families. (**A**) SH-SY5Y cell susceptibility to Ebolavirus species and entry-enhancing variant GPs. HEK293T and SH-SY5Y cells were transduced for 6 h with denoted pseudoparticles encoding for a firefly luciferase reporter gene. After 72 h, 100 µL of cell lysates were measured for luciferase activity. Unspecific entry was determined by NoEnvpp RLU values. Data are the log10 RLU mean values of 3 independent transductions with 9 individual values. Error bars represent SD. (**B**) Cell–virus specificity analysis. Huh-7.5, HEK293T, SH-SY5Y, and Jurkat cells were transduced with GFP-encoding lentiviral particles pseudotyped with different GPs or No GP (NoEnvpp) for 6 h at a MOI of 0.1 (titers determined in Huh-7.5). Seventy-two hours later, cells were analyzed for GFP expression by flow cytometry. The graph is the representation of the mean percentage of transduced cells plus individual values of the three independent experiments ± SD.

**Figure 4 viruses-11-00275-f004:**
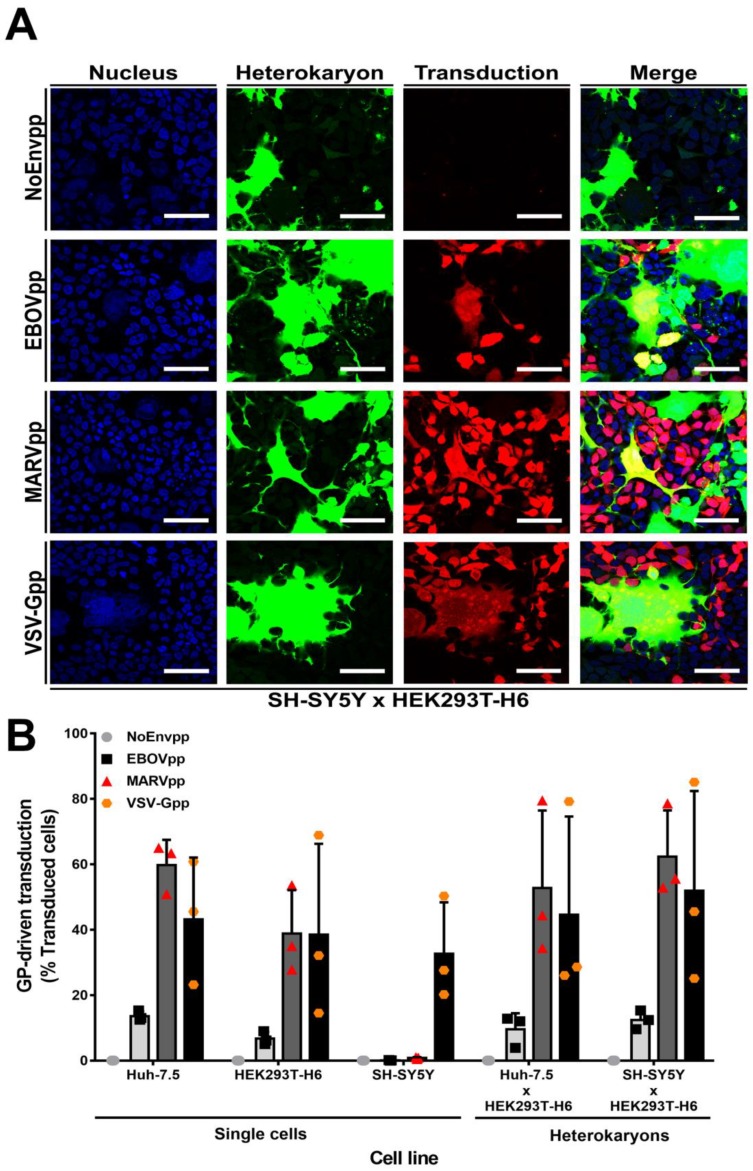
SH-SY5Y cells do not express a dominant entry restriction factor. Huh-7.5 or SH-SY5Y cells stably expressing the tetracycline-inducible Tet On 3G transactivator protein were cocultured with HEK293T-H6 cells stably expressing transactivator-inducible ZsGreen1 green fluorescent protein for 24 h. Cells were chemically fused with PEG. One hour after fusion, cells were transduced with NoEnv, EBOV, MARV, or VSV-G pseudoparticles encoding mCherry for 6 h at 37 °C. Seventy-two hours post transduction, cells were fixed with 3% PFA and analyzed for heterokaryon formation (ZsGreen1 protein expression) and susceptibility to pseudoparticle infection (mCherry protein expression). (**A**) Confocal microscopy images of one representative experiment with SH-SY5Y and HEK293T-H6 cells. Scale bars = 50 µm (**B**) Quantification of transduced cells by flow cytometry. Left-hand side of graph represents transduction percentage of single cell line controls and right-hand side from heterokaryons (discriminated previously by gating on ZsGreen1 positive cells). Mean ± SD. of three independent cell fusions and subsequent transduction experiments (*n* = 3).

**Figure 5 viruses-11-00275-f005:**
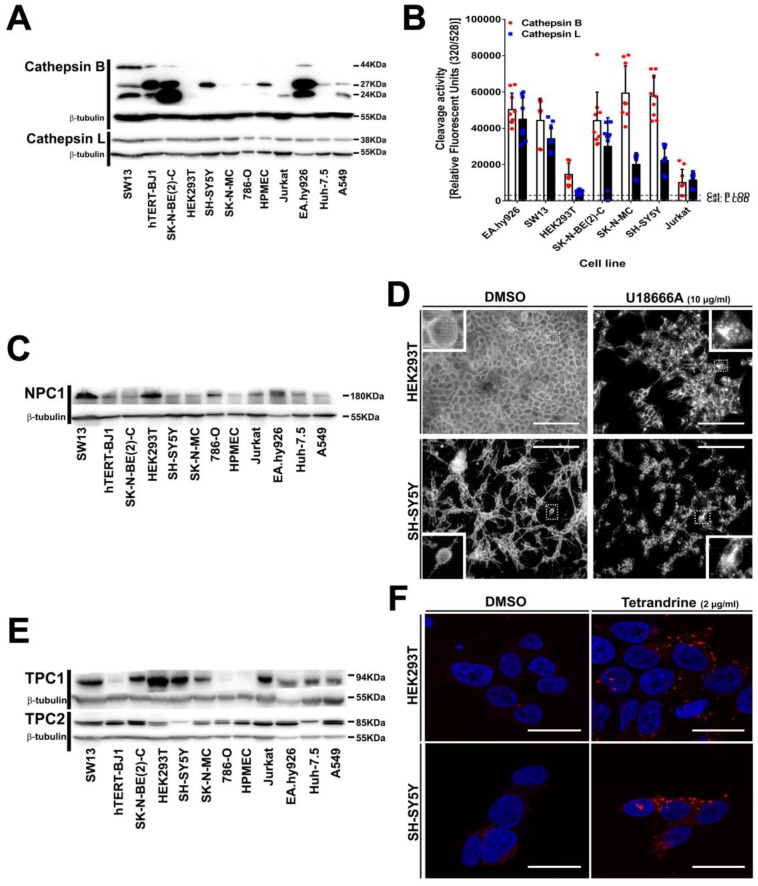
Intracellular entry factors are expressed and functional in SH-SY5Y cells. Endogenous protein expression of (**A**) cathepsin B and L, (**C**) NPC1, and (**E**) TPC1 and 2. Cell lysates were analyzed for protein expression by western blotting with protein specific abs. β-tubulin (55 kDa) was used as internal control. (**B**) Cathepsin B and L activity assay. Cathepsins substrate-specific proteolytic cleavage was measured with a commercially available kit as described in methods. Assay limit of detection (LOD) represented as a dotted line. Experiments were conducted thrice with graph bars representing mean experimental value, individual values and SD. (**D**) Intracellular cholesterol accumulation. HEK293T (upper panels) or SH-SY5Y cells (bottom panels) were treated either with vehicle (left) or with 10 µg/mL of the NPC1 inhibitor U18666A (right) for 24 h. Cells were fixed with 0.1% TX100 and stained for unesterified cholesterol using filipin. Images are representative of two independent experiments. Scale bar 200 µm. (**F**) EGF intracellular accumulation. HEK293T (upper panels) or SH-SY5Y cells (bottom panels) were treated either with vehicle alone (left) or with the TPC1/2 inhibitor tetrandrine (2 µg/mL) (right) for 24 h followed by incubation with EGF-Alexa Fluor 555 (red) for 30 min. Cells were fixed with 3% PFA, permeabilized with 0.1% TX100, stained with DAPI (blue) and analyzed by confocal microscopy. Scale bar 20 µm.

**Figure 6 viruses-11-00275-f006:**
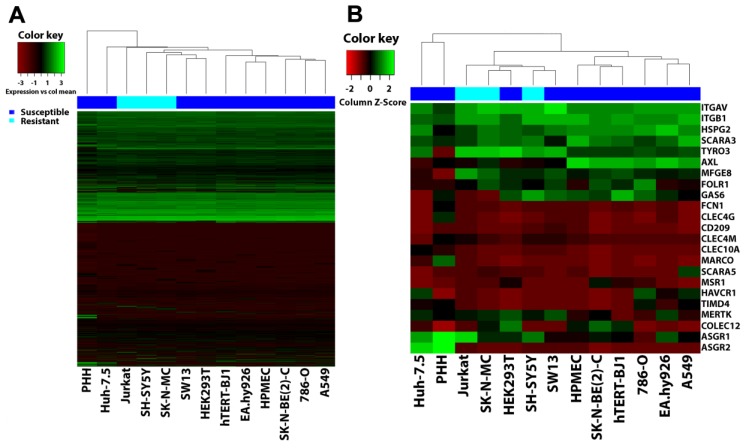
Hierarchical clustering analysis (HCA) does not correlate gene expression to susceptibility to filovirus infection. Microarray data of cell lines tested for susceptibility to filovirus infection as well as primary human hepatocytes (PHH) were clustered based on (**A**) their global gene expression or (**B**) gene expression of attachment factors implicated in filovirus entry using the heatmaps.2 of the R library “glplots” package. Transcript probes that yielded no detectable signal were removed prior to analysis. Heatmaps were generated by plotting cell lines as columns and genes as rows using the “complete” method for clustering and “Euclidean” method for distance calculation. In the bar above the heatmaps dark blue represents susceptible cell lines and light blue resistant cell lines.

**Figure 7 viruses-11-00275-f007:**
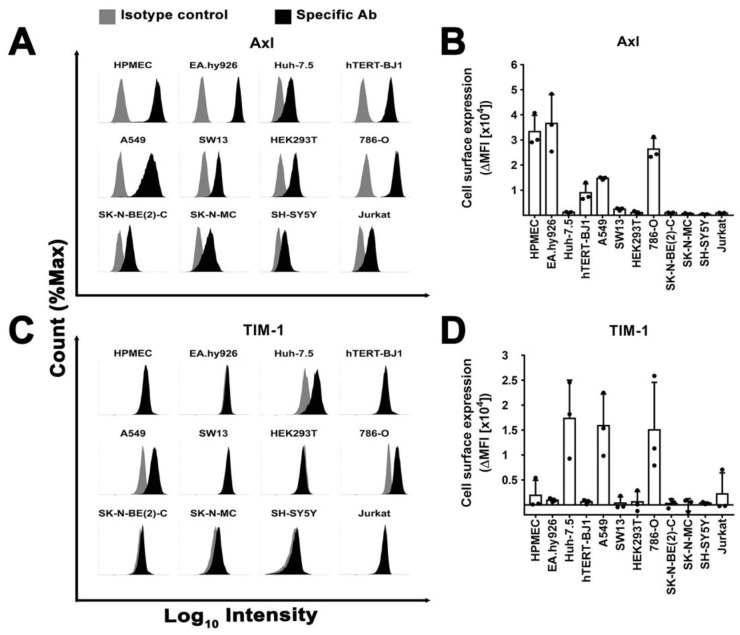
Surface expression does not explain susceptibility to filovirus infection. (**A**–**D**) Axl and TIM-1 cell surface expression. Cells were surface stained with saturating concentrations of specific abs and their correspondent isotype controls and fluorescent signal quantified by flow cytometry. Protein expression profiles of (**A**) Axl and (**C**) TIM-1 in different cell lines are shown as histograms of a representative experiment. (**B**–**D**) Cell surface expression of Axl and TIM-1 in terms of average delta mean fluorescence intensity (ΔMFI) from three independent stainings. ΔMFI was calculated by subtracting the geometric mean intensity values of the isotype control from the specific staining values. SD is shown as error bars.

**Figure 8 viruses-11-00275-f008:**
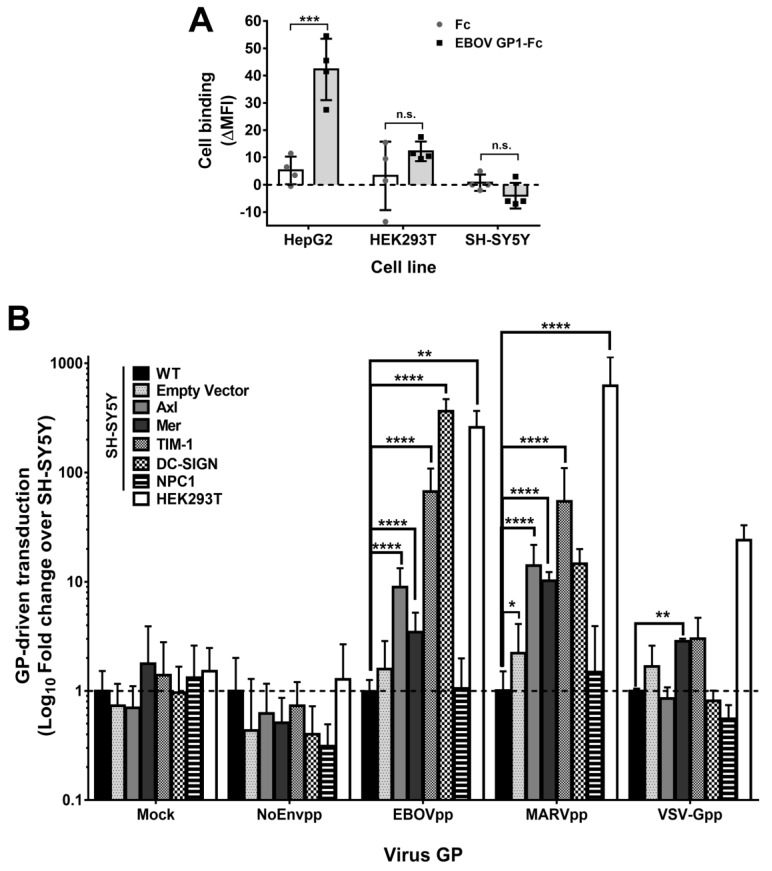
Impaired filovirus attachment on SH-SY5Y cells can be overcome by surface factor overexpression. (**A**) Binding of EBOV GP-Fc fusion protein to cell surface. EBOV-GP1 and human Fc (100 nM) were incubated for 1.5 h with the indicated cell lines and detected with a secondary ab against the human Fc fragment. MFI signal was recorded for Fc or EBOV GP1-Fc and subtracted from secondary ab MFI. Bars represent mean of two biological replicates performed in duplicate (*n* = 4) and symbols represent each individual value. Error bars represent the SD. (**B**) Overexpression of several cell surface factors and their role in filoviral GP-dependent entry. Susceptible HEK293T, resistant SH-SY5Y WT or SH-SY5Y cells genetically engineered to individually express different entry host factors were transduced with NoEnv, EBOV, MARV, and VSV-G luciferase-encoding pseudoparticles. Mock infection was performed to control background signal. Enhanced susceptibility to filovirus infection was calculated as the fold change difference over SH-SY5Y WT cells by dividing the RLU values of 100 µl lysed cells of each engineered cell line and HEK293T cells with the SH-SY5Y WT RLU values. Fold change differences between SH-SY5Y WT and HEK293T or engineered SH-SY5Y cell lines were used for statistical analysis. Graph is the representation of 3 independent transductions done in triplicate (*n* = 9). Error bars depicts SD. For the analysis of significance in all 3 graphs a multiple *t* test with a Holm–Sidak multiple comparison correction method was conducted. *P*-value significance is shown as: n.s. *P* > 0.05; * *P* ≤ 0.05; ** *P* ≤ 0.01; *** *P* ≤ 0.001; **** *P* ≤ 0.0001.

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
