# Peer review of "Characterization of the Filovirus-Resistant Cell Line SH-SY5Y Reveals Redundant Role of Cell Surface Entry Factors"

_viruses, 2019, doi:10.3390/v11030275_

Round 1
Reviewer 1 Report
The article by Zapatero-Belinchón et al. describes the neuroblastoma SH-SY5Y cell line that is specifically resistant to filovirus infection. The resistance of this adherent non-lymphocytic cell line is shown to be related to an impairment of attachment of Ebola or Marburg Envelope Glycoprotein to the cell membrane that is circumvented by over-expression of molecules such as Axl, Mer, TIM-1 or DC-SIGN, know to participate in mediating cell entry of filoviruses. The experimental work is well designed and different models to exhaustively characterize viral resistance, including full infectious Ebola and Marburg viruses, have been used. The SH-SY5Y cell line could be certainly useful to further study cell entry of filoviruses. In this respect, and to understand the cell surface requirement for infection, it would have been interesting to compare the expression of the whole spectrum, not just two, of attachment factors described for filoviruses in resistant cells, Jurkat T-lymphocytes, SH-SY5Y, versus a panel of susceptible cell lines.
Author Response
Revision of viruses-457851
Title: Characterization of the filovirus resistant cell line SH-SY5Y reveals redundant role of cell surface entry factors
Point-by-point response to reviewer 1:
We would like to thank the editor and the reviewer for their thorough and generally quite positive assessment of our work and for the helpful input they provided. Please find our specific response to the points that were raised below.
Comments and Suggestions for Authors
The article by Zapatero-Belinchón et al. describes the neuroblastoma SH-SY5Y cell line that is specifically resistant to filovirus infection. The resistance of this adherent non-lymphocytic cell line is shown to be related to an impairment of attachment of Ebola or Marburg Envelope Glycoprotein to the cell membrane that is circumvented by over-expression of molecules such as Axl, Mer, TIM-1 or DC-SIGN, know to participate in mediating cell entry of filoviruses. The experimental work is well designed and different models to exhaustively characterize viral resistance, including full infectious Ebola and Marburg viruses, have been used. The SH-SY5Y cell line could be certainly useful to further study cell entry of filoviruses. In this respect, and to understand the cell surface requirement for infection, it would have been interesting to compare the expression of the whole spectrum, not just two, of attachment factors described for filoviruses in resistant cells, Jurkat T-lymphocytes, SH-SY5Y, versus a panel of susceptible cell lines.
The decision to assess cell surface expression of Axl and TIM-1 (Figure 7) was based on the microarray analysis we performed in the panel of twelve cell lines. In Table S2, we looked into detail on the mRNA expression from all the reported filovirus attachments factors and categorized them as likely negative for transcripts below 150 Relative Light Units (RLUs), uncertain for gene transcripts in between 150-300 RLUs, and likely expressed from transcripts that yielded RLU values above 300. Those gene transcripts that score above 300 RLU or below 150 for all the cell lines were not considered for protein expression analysis. For example, DC-SIGN (CD209) was not expressed at the mRNA level in any of the cell lines. Conversely, TYRO3 mRNA was detected in all the immortalized cell lines and thus, could not be the determinant of filovirus cell tropism. Integrin αV and β1 (Figure S5) were the only exception. Even though their RLU values was above 300 in all the cell lines, they seemed to have much lower values on the resistant cell lines and therefore we checked them at the protein level.
Nevertheless, we agree that it would be interesting to see what attachment factors are expressed on which cell lines and we have added this on the discussion section. Furthermore, we have made clearer the rationale behind the selection of attachment factors for further analysis by adding the abovementioned examples (DC-SIGN and TYRO3). Finally, we have updated Table S2 to include a legend detailing the expression criteria.
Reviewer 2 Report
In this manuscript by Zapatero-Belinchó F.J. et al., the authors have identified neuroblastoma cell lines, SK-N-MC and SH-SY5Y, to be resistant to EBOV and MARV infection. By fusion to susceptible cells, they have determined that the lack of infection in these cells was not due to the expression of a restriction factor, but rather to the absence of at least one cellular factor required for entry. They analyzed the expression of the known triggering factors, as well as their activity in some instance in susceptible and non-susceptible cells, and did not find any correlation with resistance suggesting that other entry factors are at play. Using a soluble EBOV GP1 construct, they found that surface binding was impaired in the resistant cells, also binding to the susceptible 293T cells was also negligible. While analysis of the surface expression of different known attachment factors seemed to point out to an overall lower expression in the non-susceptible cells, the correlation was weak when compared to the expression in susceptible cells. Although the identity of the lacking entry factor(s) in the resistant cells was/were not found, the expression of any of the known filovirus attachment factor restored susceptibility.
Overall, the study is well-conducted and the paper well-written. I only have a few specific concerns:
Lines 705-706: The authors state: “In sum, expression of TIM-1, Axl, Mer or DC-SIGN is not necessary but sufficient to confer susceptibility to filoviruses.” This statement if misleading, as it is incorrect to say that their expression is sufficient as the triggering factors (Cathepsins and NPC1) are the ones that are absolutely required. Without these, overexpression of TIM-1, Axl, Mer and DC-SIGN alone or even in combination would not render any cells susceptible. I strongly suggest to refrain from using sufficient, but would use: important determinant of susceptibility to filoviruses.
While an effect on GP binding to cells was observed, virus binding, which can occur via envelope phosphatidylserine interactions with the host cell, was not assessed. Also, a defect in macropinocytosis internalization was not investigated (which is understandable due to the decrease binding that would complicate analysis). However, while expression of the different attachment factors did increase significantly GP-mediated entry, surface binding in these conditions was not measured. This means that while they did indeed confer susceptibility, the mechanism by which this occurred is unknown and could be different for each. For instance, TIM-1 and DC-SIGN could increase binding, while Axl and Mer could enhance macropinocytosis, as ectopic expression of axl was previously shown to increase macropinocytic uptake. This could potentially explain the difference in the folds increase that was observed. This possibility should at least be proposed in the discussion, maybe in the paragraph of lines 716-736.
Minor:
Line 51, remove the s at the end of infect
Remove (Figure. 5E) in line 252.
Author Response
Revision of viruses-457851
Title: Characterization of the filovirus resistant cell line SH-SY5Y reveals redundant role of cell surface entry factors
Point-by-point response to reviewer 2:
We would like to thank the editor and the reviewer for their thorough and generally quite positive assessment of our work and for the helpful input they provided. Please find our specific response to the points that were raised below.
Comments and Suggestions for Authors
In this manuscript by Zapatero-Belinchón F.J. et al., the authors have identified neuroblastoma cell lines, SK-N-MC and SH-SY5Y, to be resistant to EBOV and MARV infection. By fusion to susceptible cells, they have determined that the lack of infection in these cells was not due to the expression of a restriction factor, but rather to the absence of at least one cellular factor required for entry. They analyzed the expression of the known triggering factors, as well as their activity in some instance in susceptible and non-susceptible cells, and did not find any correlation with resistance suggesting that other entry factors are at play. Using a soluble EBOV GP1 construct, they found that surface binding was impaired in the resistant cells, also binding to the susceptible 293T cells was also negligible. While analysis of the surface expression of different known attachment factors seemed to point out to an overall lower expression in the non-susceptible cells, the correlation was weak when compared to the expression in susceptible cells. Although the identity of the lacking entry factor(s) in the resistant cells was/were not found, the expression of any of the known filovirus attachment factor restored susceptibility.
Overall, the study is well-conducted and the paper well-written. I only have a few specific concerns:
· Point 1: Lines 705-706: The authors state: “In sum, expression of TIM-1, Axl, Mer or DC-SIGN is not necessary but sufficient to confer susceptibility to filoviruses.” This statement if misleading, as it is incorrect to say that their expression is sufficient as the triggering factors (Cathepsins and NPC1) are the ones that are absolutely required. Without these, overexpression of TIM-1, Axl, Mer and DC-SIGN alone or even in combination would not render any cells susceptible. I strongly suggest to refrain from using sufficient, but would use: important determinant of susceptibility to filoviruses.
We agree with the suggestion and the sentence has been changed accordingly.
· Point 2: While an effect on GP binding to cells was observed, virus binding, which can occur via envelope phosphatidylserine interactions with the host cell, was not assessed. Also, a defect in macropinocytosis internalization was not investigated (which is understandable due to the decrease binding that would complicate analysis). However, while expression of the different attachment factors did increase significantly GP-mediated entry, surface binding in these conditions was not measured. This means that while they did indeed confer susceptibility, the mechanism by which this occurred is unknown and could be different for each. For instance, TIM-1 and DC-SIGN could increase binding, while Axl and Mer could enhance macropinocytosis, as ectopic expression of axl was previously shown to increase macropinocytic uptake. This could potentially explain the difference in the folds increase that was observed. This possibility should at least be proposed in the discussion, maybe in the paragraph of lines 716-736.
This is a very valid point that we have added to the discussion section as suggested.
Minor:
· Point 3: Line 51, remove the s at the end of infect
Done
· Point 4: Remove (Figure. 5E) in line 252.
Done
Reviewer 3 Report
The manuscript by Zapatero-Belinchon describes the identification of attachment factors, which play an important role in limiting filovirus entry. In a series of well designed and executed studies, the authors demonstrate that plasma membrane factors mediate attachment of filoviruses to host cells and subsequently are involved in facilitating infection of target cells.
The data are very clearly presented and easy to follow. There are only a few minor comments to address:
Lines 51-52: Better expand to rodents (instead of mice and guinea pigs, to cover hamsters as well) and include ferrets as well.
Line 392: Exchange "challenge" with "infection".
Line 432: Results for lentiviral particles carrying SARS and LASV glycoproteins are not described.
Have the authors considered to generate heterokaryons between Jurkat cells and susceptible cells? This would have been a nice addition to this work.
Figure 5 is difficult to read and too small.Figure 6: The labels at the plots are small and difficult to read.
Lines 688, 711, and 718: References need to be reformatted.
Line 698: Spell out "abs" as antibodies.
Author Response
Revision of viruses-457851
Title: Characterization of the filovirus resistant cell line SH-SY5Y reveals redundant role of cell surface entry factors
Point-by-point response to reviewer 3:
We would like to thank the editor and the reviewer for their thorough and generally quite positive assessment of our work and for the helpful input they provided. Please find our specific response to the points that were raised below.
Comments and Suggestions for Authors
The manuscript by Zapatero-Belinchon describes the identification of attachment factors, which play an important role in limiting filovirus entry. In a series of well designed and executed studies, the authors demonstrate that plasma membrane factors mediate attachment of filoviruses to host cells and subsequently are involved in facilitating infection of target cells.
The data are very clearly presented and easy to follow. There are only a few minor comments to address:
· Point 1: Lines 51-52: Better expand to rodents (instead of mice and guinea pigs, to cover hamsters as well) and include ferrets as well.
Done.
· Point 2: Line 392: Exchange "challenge" with "infection".
Done.
· Point 3: Line 432: Results for lentiviral particles carrying SARS and LASV glycoproteins are not described.
We agree there may not have been sufficient information regarding these two GPs and we have added information on the appropriate results section accordingly.
· Point 4: Have the authors considered to generate heterokaryons between Jurkat cells and susceptible cells? This would have been a nice addition to this work.
We have indeed considered to fuse Jurkat cells with susceptible cells such as HEK293T or even with SH-SY5Y cells overexpressing any of the attachment factors. This possibility is being discussed in lines 737-739. Since the present work is focused on SH-SY5Y cells we feel that it may be outside the scope of the paper. We are however very interested to perform these experiments in the future in order to better understand the mechanism underlying the resistant nature of Jurkat cells.
· Point 5: Figure 5 is difficult to read and too small
We have reformatted the figure in vertical to maximize the quality of the image and increase legibility.
· Figure 6: The labels at the plots are small and difficult to read.
We have increased the font size of the labels to make them easy to read.
· Point 7: Lines 688, 711, and 718: References need to be reformatted.
Done.
· Point 8: Line 698: Spell out "abs" as antibodies.
Done.